# ECG pathology and its association with death in critically ill COVID-19 patients, a cohort study

Jacob Rosén[1]*, Maria Noreland[2], Karl Stattin[1], Miklós Lipcsey[1,3], Robert Frithiof[1], Andrei Malinovschi[2‡], Michael Hultström[1,4‡], on behalf of the Uppsala Intensive Care COVID-19 Research Group[¶]

1 Department of Surgical Sciences, Anaesthesiology and Intensive Care Medicine, Uppsala University, Uppsala, Sweden, 2 Department of Medical Sciences, Clinical Physiology, Uppsala University, Uppsala, Sweden, 3 Hedenstierna laboratory, CIRRUS, Department of Surgical Sciences, Anaesthesiology and Intensive Care Medicine, Uppsala University, Uppsala, Sweden, 4 Department of Medical Cell Biology, Integrative Physiology, Uppsala University, Uppsala, Sweden

‡ AM and MH are joint senior authors.
¶ Membership of the Uppsala Intensive Care COVID-19 Research Group is listed in the Acknowledgments.
* jacob.rosen@surgsci.uu.se

**Data Availability Statement:** Data privacy regulations prohibit deposition of individual level data to public repositories and the ethical approval does not cover public sharing of data for unknown

## Abstract

### Background

We investigated the prevalence of ECG abnormalities and their association with mortality, organ dysfunction and cardiac biomarkers in a cohort of COVID-19 patients admitted to the intensive care unit (ICU).

### Methods

This cohort study included patients with COVID-19 admitted to the ICU of a tertiary hospital in Sweden. ECG, clinical data and laboratory findings during ICU stay were extracted from medical records and ECGs obtained near ICU admission were reviewed by two independent physicians.

### Results

Eighty patients had an acceptable ECG near ICU-admission. In the entire cohort 30-day mortality was 28%. Compared to patients with normal ECG, among whom 30-day mortality was 16%, patients with ECG fulfilling criteria for prior myocardial infarction had higher mortality, 63%, odds ratio (OR) 9.61 (95% confidence interval (CI) 2.02–55.6) adjusted for Simplified Acute Physiology Score 3 and patients with ST-T abnormalities had 50% mortality and OR 6.05 (95% CI 1.82–21.3) in univariable analysis. Both prior myocardial infarction pattern and ST-T pathology were associated with need for vasoactive treatment and higher peak plasma levels of troponin-I, NT-pro-BNP (N-terminal pro-Brain Natriuretic Peptide), and lactate during ICU stay compared to patients with normal ECG.

purposes. Upon contact with the authors or SciLifeLab (https://doi.org/10.17044/scilifelab.14229410.v1) an institutional data transfer agreement may be established, and data shared if the aims of data use are covered by ethical approval and patient consent.

**Funding:** The study was funded by grants from SciLifeLab/KAW national COVID-19 research program project to M.H. (KAW 2020.0182 and KAW 2020.0241), the Swedish Heart-Lung Foundation (20210089, 20190639, 20190637) to M.H., the Swedish Research Council to R.F. (2014-02569 and 2014-07606) and Swedish Kidney Foundation (F2020-0054) to R.F. The funders had no role in study design, data collection and analysis, decision to publish, or preparation of the manuscript.

**Competing interests:** The authors have declared that no competing interests exist.

## Conclusion

ECG with prior myocardial infarction pattern or acute ST-T pathology at ICU admission is associated with death, need for vasoactive treatment and higher levels of biomarkers of cardiac damage and strain in severely ill COVID-19 patients, and should alert clinicians to a poor prognosis.

## Introduction

Although Corona virus disease 2019 (COVID-19) primarily affects the respiratory system and may cause severe pneumonia and hypoxemic respiratory failure [1], other organ systems are also frequently affected, such as the renal [2] and the cardiovascular system [3]. Typical cardiovascular risk factors, such as hypertension, diabetes and cerebrovascular disease are associated with a higher risk of severe disease [4, 5]. Elevated cardiac biomarkers during hospitalization due to COVID-19 are associated with higher risk of severe disease and mortality [6–11]. Proposed mechanisms for cardiac injury include demand ischemia in patients with pre-existing coronary artery stenosis, diffuse myocardial injury due to hypoxemia or arrythmia, Takotsubo syndrome, myocarditis and pulmonary hypertension secondary to adult respiratory distress syndrome (ARDS) or pulmonary embolism [3, 12–14].

The electrocardiogram (ECG) is a readily available, non-invasive, radiation free diagnostic mainstay of cardiac pathology [15] used at some point in nearly all severely ill patients [16]. ECG abnormalities in patients with COVID-19 were described in small case series early during the pandemic [17–20] and analyses of larger cohorts found that ECG pathology at hospital admission is associated with more severe disease and higher mortality [9, 21–24]. Unfortunately, previous studies have not specifically investigated patients admitted to the intensive care unit (ICU), who have the highest mortality [5]. ECG at ICU admission may differ from ECG at hospital admission and further knowledge of ECG abnormalities at ICU admission may add important diagnostic and prognostic information.

The primary aim of this study was to describe the prevalence of ECG pathology among COVID-19 patients at ICU admission and its association with mortality. We further aimed to compare laboratory findings including cardiac biomarkers in patients with and without ECG pathology.

## Methods

### Ethics statement

The protocol was approved by the National Ethical Review Agency, Uppsala, Sweden (EPM; 2020–01623), and registered at ClinicalTrials (NCT04316884). Written informed consent for access to electronic medical records was obtained from all participants, or next of kin if the patient was unable to give consent in accordance with the decision from the ethics committee. Data was acquired from electronic medical records and pseudonymised after collection. The study was performed in compliance with the Declaration of Helsinki and STROBE guidelines were followed for reporting.

### Study design, setting and population

This study was performed as a subgroup analysis of the PronMed cohort study [25] in patients consecutively admitted to a mixed medical and surgical ICU between March 23 and July 14, 2020 at Uppsala University Hospital, a tertiary care teaching hospital in Sweden.

Patients >18 years old admitted to the ICU with COVID-19 diagnosis confirmed by positive SARS-CoV-2 reverse transcription polymerase chain reaction tests on naso- or oropharyngeal swabs were eligible for inclusion in this study if they had an ECG recorded within 48 hours prior to, or within 72 hours after ICU admission. Patients were admitted to ICU based on the clinical judgement of the attending ICU physician. The main criterium was need for organ-support, most commonly respiratory support beyond high-flow nasal oxygen with 60% oxygen at 60L min$^{-1}$, or high risk of progression based on clinical judgement.

## Data collection and ECG interpretation

Demographic data, medical history, laboratory findings, treatment measures and ECG were extracted from the medical records during ICU stay. If a patient had several ECGs recorded within the predefined time limit, the ECG recorded closest to ICU-admission was chosen for analysis. Two physicians with profound experience of ECG interpretation (JR, MN) analysed the ECGs independently according to pre-specified criteria [26–29] and recorded findings in a standardized case report form. The physicians were blinded to patient outcomes during ECG interpretation. Discrepancies were resolved by consensus.

ECG data [30] included rhythm, premature atrial and ventricular contractions (PAC and PVC), frontal plane axis, PQ-time, QRS-duration, Bazett-corrected [31] QT-interval (QTc), poor R-wave progression, P-wave pathology, left and right ventricular hypertrophy (LVH and RVH), $S_1Q_3T_3$ pattern, conduction blocks, ST-segment abnormality (depression or elevation), T-wave inversion and right ventricular strain pattern.

We defined and focused the analysis on two composite ECG patterns to represent chronic and acute cardiac conditions: ECG with prior myocardial infarction (MI) pattern and ECG with ST-T pathology. ECG with prior myocardial infarction (MI) pattern was defined as presence of pathological Q-waves [32] and/or poor precordial R-wave progression [33, 34]. We did not classify poor R-wave progression as myocardial infarction if it was clearly due to LVH, RVH or conduction blocks [33]. ECG with ST-T pathology was defined as presence of pathological ST-elevation, ST-depression or T-wave inversion [27]. Laboratory data was compared in patients with either composite ECG patterns to those with normal ECG.

## Statistical analysis

All data was analysed using Microsoft Excel (Redmond, WA, USA) and the R package"rcmdr" (Rcmdr: R Commander. R package version 3.6.3). Data were presented as mean and standard deviations (SD) or median and interquartile range (IQR) for normally and non-normally distributed data respectively. Categorical variables were presented as numbers (percentages). Univariable logistic regression was performed to investigate the relationship between ECG pathology and mortality. Multivariable analyses were performed for composite ECG pathology. Due to the limited sample size and events, the multivariable analyses were adjusted for Simplified Acute Physiology Score (SAPS 3) [35] only. SAPS 3 was chosen to adjust for baseline differences, as it is a validated scoring system used for prediction of hospital mortality. Uni- and multivariable logistic regression were further performed for composite ECG pathologies for patients who had an ECG recorded within 24 hours from ICU-admission. This sensitivity analysis is presented in the Supporting information. Normally and non-normally distributed continuous data was compared using independent t-test and Mann-Whitney U-test respectively and Chi-square or Fischer's exact tests were used to compare categorical data as appropriate. Two-sided p-values <0.05 were considered statistically significant. Due to the exploratory nature of this study, we did not adjust for multiple statistical testing.

# Results

## Patient characteristics

Between March 23 and July 14, 2020, 168 patients were admitted to the ICU with confirmed COVID-19, of which 123 patients (76%) were included in the PronMed cohort. 83 patients had an ECG recorded within 48 hours prior to up to 72 hours following ICU admission. Three patients were excluded due to poor ECG quality, leaving 80 patients (65% of included patients) for final analysis (Fig 1). The mean age was 60.6 (SD 13.6), mean body mass index 30.0 kg m$^{-2}$ (SD 5.9) and 20 patients (25%) were female (Table 1). The most common comorbidities were hypertension (54%) and diabetes (26%). Ten patients (13%) had pre-existing ischemic heart disease, 14 patients (18%) had atherosclerotic vascular disease and three patients (4%) had heart failure. Patients had a mean SAPS 3 of 53 (SD 10) and a median $PaO_2/FiO_2$ ratio of 17.8 (IQR 15.5–23.5) kPa at ICU admission. Patients with an ECG pattern of prior MI and ST-T pathology were older and had higher prevalence of hypertension, diabetes, cardiovascular and vascular disease than patients with normal ECG.

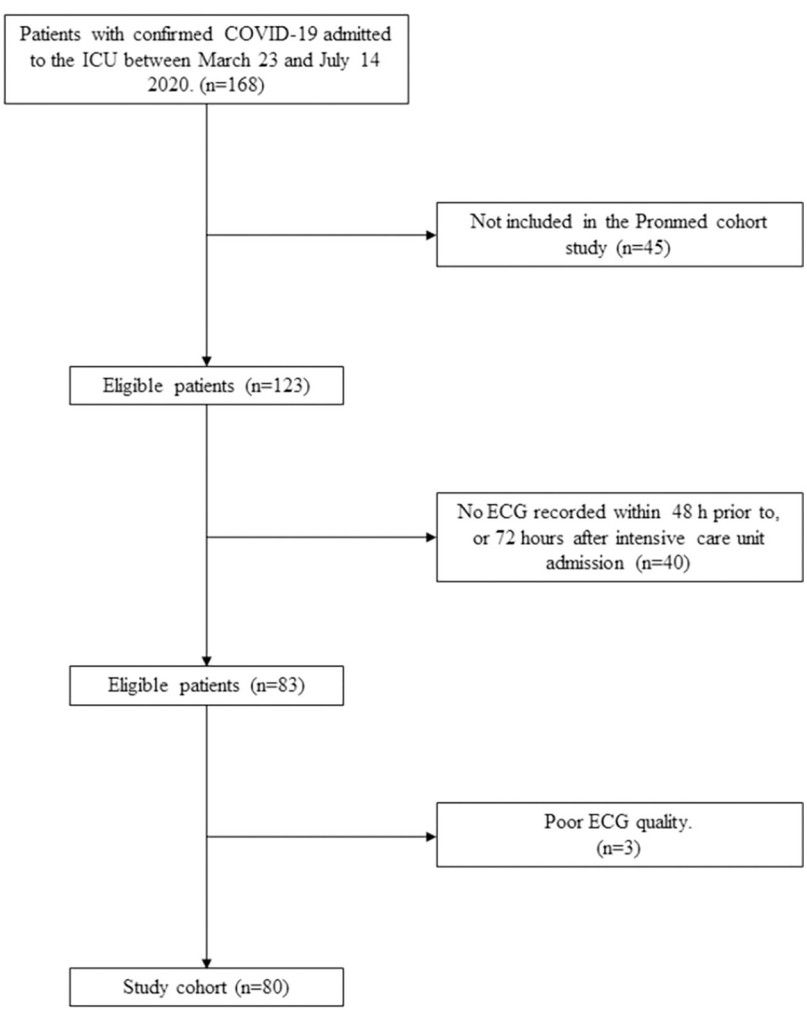

**Fig 1. Flow chart of included patients.**

**Table 1. Patient characteristics at ICU admission.**

|  | All patients (n = 80) | Normal ECG (n = 51) | Prior MI pattern (n = 11) | ST-T pathology (n = 17) |
|---|---|---|---|---|
| **Anthropometry** |  |  |  |  |
| Age (years) | 61 (14) | 56 (12) | 66 (11) | 71.5 (13.3) |
| Sex (female) | 20 (25%) | 14 (27%) | 2 (18%) | 6 (35%) |
| Weight (kg) | 89 (20) | 90 (20) | 90 (16) | 85 (20) |
| Length (cm) | 175 (9) | 175 (10) | 175 (10) | 170 (7) |
| BMI (kg m$^{-2}$) | 30 (6) | 29 (6) | 32 (6) | 31 (6) |
| **Pre-existing comorbidities** |  |  |  |  |
| Hypertension | 43 (54%) | 25 (49%) | 8 (73%) | 13 (76%) |
| Diabetes Mellitus | 21 (26%) | 10 (20%) | 6 (55%) | 7 (41%) |
| Ischemic heart disease | 10 (13%) | 3 (6%) | 4 (36%) | 3 (18%) |
| Heart failure | 3 (4%) | 1 (2%) | 2 (18%) | 2 (12%) |
| Pulmonary embolism | 1 (1%) | 0 (0%) | 1 (9%) | 1 (6%) |
| Pulmonary hypertension | 0 (0%) | 0 (0%) | 0 (0%) | 0 (0%) |
| Malignant disease | 7 (9%) | 2 (4%) | 0 (0%) | 5 (29%) |
| Liver failure | 1 (1%) | 1 (2%) | 0 (0%) | 0 (0%) |
| Pulmonary disease | 22 (28%) | 16 (31%) | 1 (9%) | 5 (29%) |
| Vessel disease | 14 (18%) | 4 (8%) | 6 (55%) | 4 (24%) |
| **ICU characteristics** |  |  |  |  |
| Days with symptoms at ICU-admission | 8.5 (5–16) | 8.5 (5–15) | 16 (9–18) | 7 (6–12) |
| SAPS 3 | 53 (10) | 51 (8) | 54 (7) | 63 (8) |
| Average PaO$_2$/FiO$_2$ ratio ICU-day 1 (kPa) | 17.7 (15.5–23.5) | 19.0 (15.7–24.3) | 16.1 (15.1–18.4) | 16.0 (15.1–18.4) |
| Pulmonary embolism diagnosed during ICU stay | 9 (11%) | 7 (13%) | 2 (18%) | 0 (0%) |

Data are presented as mean (standard deviation), median (interquartile range) and numbers (percentages). ICU: Intensive care unit. MI: myocardial infarction. SAPS 3: Simplified Acute Physiology Score 3 [35].

## ECG characteristics

ECG were recorded at median day 0 (IQR -1 to 0) of ICU stay. The majority of patients were in sinus rhythm at ICU admission (Table 2) and 51 (64%) had a normal ECG. The mean heart rate was 89 (SD 17) and 24 patients were tachycardic (defined as a heart rate $\geq$100 min$^{-1}$). Eleven patients (14%) had an ECG pattern consistent with prior MI although only four of these had a prior diagnosis of ischemic heart disease. 17 patients (21%) had an ECG with ST-T-pathology. Twenty-seven (34%) patients had a conduction block but LBBB and RBBB were rare. T-wave inversion (n = 13, 16%) was the most common repolarization abnormalities, whereas ST-depression or ST-elevation were found in six patients (8%). Although RBBB or S$_1$Q$_3$T$_3$ morphology were noticed in isolation in three patients, there were no patients with an ECG compatible with acute right ventricular strain and none of the 9 patients (11%) who were diagnosed with pulmonary embolism prior to or following ICU admission (Table 1) had ECG characteristics indicative of the diagnosis.

## Mortality

At 30 days follow-up, 22 patients (28%) had died. Among patients with normal ECG, 30-day mortality was 16%. Among patients with an ECG consistent with prior MI pattern mortality was 63% and among patients with ST-T pathology, mortality was 50%. In univariable logistic regression analysis, prior MI pattern, ST-T pathology, PAC and/or PVC, pathological Q-wave, poor R-wave progression, ST-depression and T-wave inversion were associated with higher

**Table 2. ECG characteristics within 48 hours prior to, or up to 72 hours after, admission to the intensive care unit.**

| ECG characteristics | All patients (n = 80) |
|---|---|
| *Composite EGG patterns* | |
| Normal ECG | 51 (64%) |
| Prior myocardial infarction | 11 (14%) |
| ST-T-pathology | 17 (21%) |
| *Rhythm* | |
| Sinus rhythm | 76 (95%) |
| Heart rate | 89 (17) |
| Atrial fibrillation | 3 (4%) |
| Heart rate (range) | (100–178) |
| SVT | 1 (1%) |
| Tachycardia (HR>100) | 24 (30%) |
| Bradycardia (HR<50) | 1 (1.3%) |
| *Premature contraction* | |
| Atrial | 4 (5%) |
| Ventricular | 5 (6%) |
| *ECG measurements* | |
| QRS-axis (°) | 14 (39) |
| Left axis deviation (<-30°) | 7 (9%) |
| Right axis deviation (>+90°) | 2 (3%) |
| PR interval (ms) | 153 (24) |
| QRS duration (ms) | 91 (16) |
| QRS > 120 ms n(%) | 2 (3%) |
| QTc (ms) | 440 (27) |
| QTc>500 ms | 2 (3%) |
| *Conduction blocks* | |
| AV-block I, II or III | 0 (0%) |
| Intraventricular block | 14 (18%) |
| Left hemiblock | 5 (6%) |
| Partial RBBB | 4 (5%) |
| Partial LBBB | 2 (3%) |
| RBBB | 1 (1%) |
| LBBB | 1 (1%) |
| *Morphology* | |
| Early repolarisation | 2 (3%) |
| Pathological Q-wave | 7 (9%) |
| Poor R-wave progression | 5 (6%) |
| $S_1Q_3T_3$-morphology | 2 (3%) |
| Left ventricular hypertrophy | 6 (8%) |
| ST-elevation | 1 (1%) |
| ST-depression | 5 (6%) |
| T-wave inversion | 13 (16%) |
| U-wave | 1 (1%) |

Data are presented as mean (standard deviation) or n (%) if not stated otherwise. Prior myocardial infarction includes ECG with Q-wave and/or poor R-wave progression. ST-T-pathology includes ECG with ST-elevation, ST-depression or T-wave inversion. SVT: supraventricular tachycardia, QTc: Corrected QT interval according to Bazett. RBBB: Right bundle branch block. LBBB: Left bundle branch block.

**Table 3. Analysis of odds ratio for death within 30 days of intensive care unit admission.**

| ECG-abnormalities | Day of ECG recording[a] median (IQR) | Survived (n = 58) | Died (n = 22) | Univariable analysis | | Multivariable analysis | |
|---|---|---|---|---|---|---|---|
| | | | | OR (95% CI) | P-value | OR (95% CI) | P-value |
| **Composite abnormalities** | | | | | | | |
| Normal ECG | 0 (-1, 1) | 43 (74%) | 8 (36%) | Ref. | n.a. | Ref. | n.a. |
| Prior MI pattern | 0 (-1, 1) | 4 (7%) | 7 (31%) | 9.63 (2.38–44.8) | 0.002 | 9.61 (2.02–55.6) | 0.006 |
| ST-T pathology | 0 (0, 0) | 8 (14%) | 9 (41%) | 6.05 (1.82–21.3) | 0.004 | 1.95 (0.42–8.52) | 0.38 |
| **Single abnormalities** | | | | | | | |
| Heart rate (min$^{-1}$) | n.a. | 91 (20) | 92 (21) | 1.00 (0.98–1.03) | 0.86 | | |
| QTc (ms) | n.a. | 441 (27) | 437 (27) | 0.99 (0.97–1.01) | 0.57 | | |
| Tachycardia (>100 min$^{-1}$) | 0 (-1, 0) | 16 (28%) | 8 (36%) | 3.30 (0.95–12.5) | 0.065 | | |
| PAC and/or PVC | 0 (0, 1) | 4 (7%) | 5 (23%) | 7.32 (1.57–36.9) | 0.011 | | |
| Conduction block | 0 (-1, 1) | 21 (36%) | 6 (27%) | 1.18 (0.33–4.07) | 0.79 | | |
| LVH | 0 (0, 0) | 3 (5%) | 3 (14%) | 5.50 (0.86–34.8) | 0.059 | | |
| Q-wave | -1 (-1, 1) | 3 (5%) | 4 (18%) | 7.33 (1.38–43.8) | 0.020 | | |
| Poor R-wave progression | 0 (0, 0) | 1 (2%) | 4 (18%) | 22.0 (2.82–462) | 0.009 | | |
| ST-depression | 0 (0, 1) | 2 (3%) | 3 (14%) | 8.25 (1.19–70.6) | 0.033 | | |
| T-wave inversion | 0 (0–1) | 6 (10%) | 7 (31%) | 6.42 (1.73–25.4) | 0.006 | | |

Data are presented as absolute numbers (percentages), mean (standard deviation) or median (IQR). Logistic regression analysis was performed with normal ECG (n = 51) as reference. Multivariable analysis performed for composite ECG patterns and adjusted for SAPS 3. CI: Confidence interval. SAPS 3: Simplified Acute Physiology Score [35]. Prior MI (myocardial infarction) pattern includes ECG with Q-wave and/or poor R-wave progression. ST-T-pathology includes ECG with ST-elevation, ST-depression or T-wave inversion. PAC: premature atrial contraction, PVC: premature ventricular contraction. QTc = Corrected QT time according to Bazett.

[a]Day zero defined as the day of intensive care unit admission.

odds of death (Table 3). In multivariable analysis for the composite ECG-patterns, adjusting for SAPS 3, prior MI pattern was associated with higher odds of death. Kaplan-Meier survival analyses in patients with normal ECG versus patients with prior MI pattern or ST-T pathology are presented in Figs 2 and 3, respectively. Patients with ECG-abnormalities had their ECG recorded at similar time in relation to ICU admission compared with patients who had a normal ECG. Analysis of ECGs recorded within 24 hours from ICU admission (n = 60) did not differ from the main analysis (S1 Appendix).

## Biomarkers and organ dysfunction

Patients with prior MI pattern or ST-T-pathology at ICU admission had higher peak plasma values during ICU stay of troponin-I, NT-pro-BNP (N-terminal pro-Brain Natriuretic Peptide) and lactate compared to patients who had normal ECGs, but similar markers of inflammation (Table 4). More patients with prior MI pattern required treatment with vasoactive drugs compared to patients with normal ECG. Peak plasma levels of creatinine were higher in patients with prior MI pattern and higher but not statistically significant among patients with ST-T pathology compared to patients with normal ECG.

## Discussion

In this cohort study of 80 critically ill COVID-19 patients, several ECG pathologies were associated with death in the univariable analysis, including both prior MI pattern and ST-T pathology. An ECG consistent with prior MI pattern was associated with death in multivariable analysis adjusting for SAPS 3. Patients with prior MI pattern probably represents a population

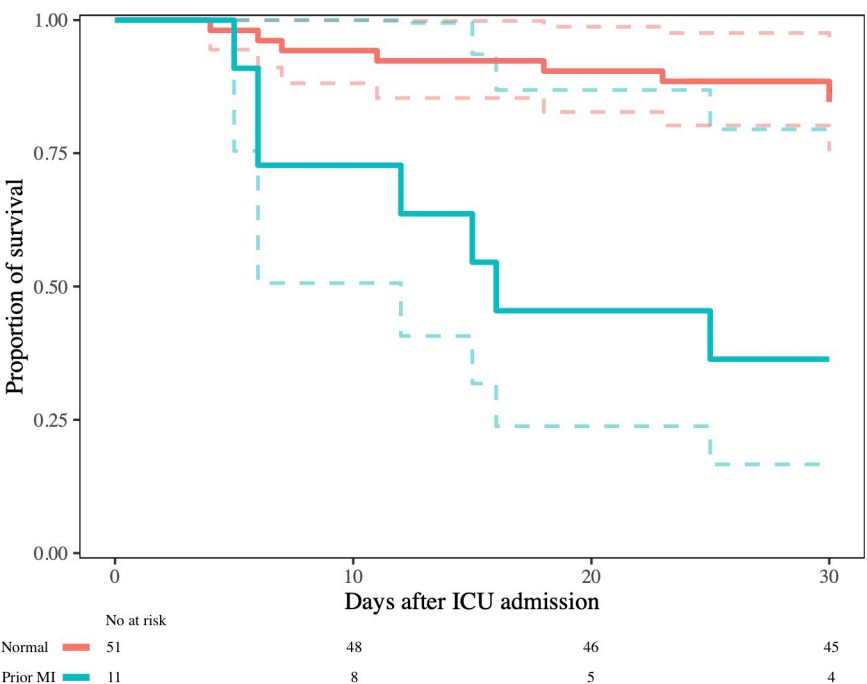

**Fig 2. Kaplan-Meier survival analysis.** Comparison of patients with normal ECG and patients with prior myocardial infarction pattern (pathological Q-waves and/or poor R-wave progression), log-rank test p<0.001.

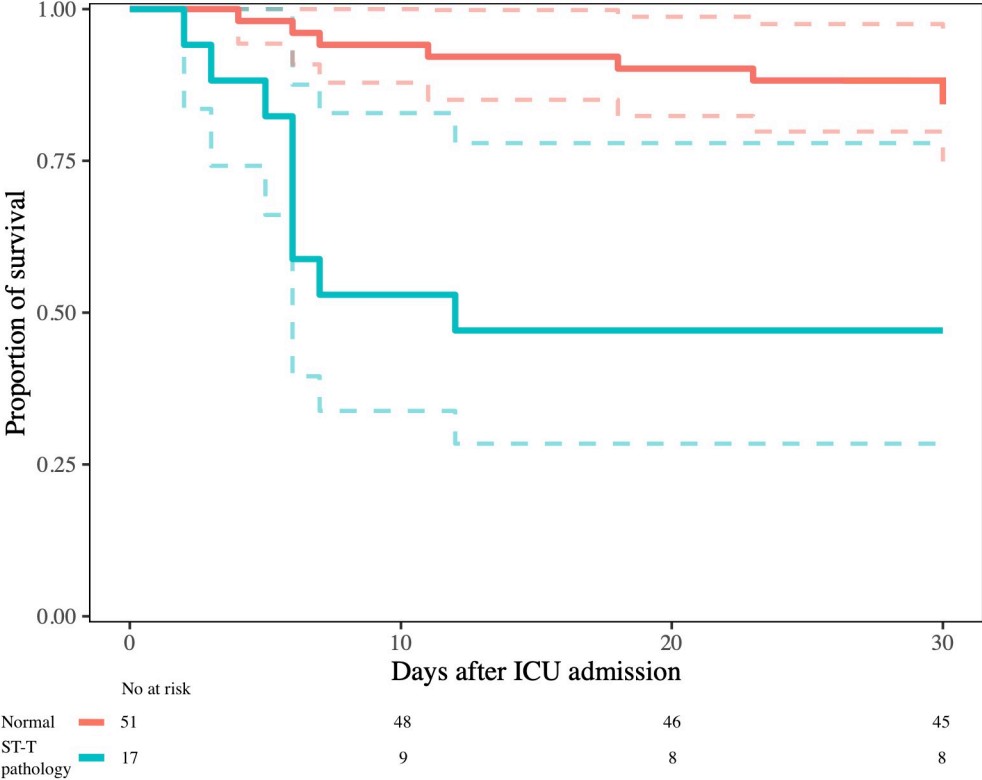

**Fig 3. Kaplan-Meier survival analysis.** Comparison of patients with normal ECG and patients with ST-T pathology (ST-elevation, ST-depression or T-wave inversion), log-rank test p<0.001.

**Table 4. Comparison of peak plasma laboratory values and organ support during intensive care unit stay between patients with normal ECG (reference) and ECG with prior myocardial infarction pattern or ST-T pathology respectively at intensive care unit admission.**

|  | Normal ECG (n = 51) | Prior MI pattern (n = 11) | P-value | ST-T pathology (n = 17) | P-value |
|---|---|---|---|---|---|
| CRP | 281 (166–378) | 294 (251–375) | 0.34 | 261 (222–352) | 0.99 |
| IL-6 | 118 (32–342) | 301 (181–369) | 0.11 | 260 (121–458) | 0.52 |
| Ferritin | 2587 (831–3913) | 1786 (1160–3954) | 0.84 | 2253 (821–5145) | 0.89 |
| Troponin-I | 15 (7–26) | 104 (32–189) | 0.003 | 116 (75–293) | <0.001 |
| NT-pro-BNP | 555 (274–1210) | 3390 (529–5480) | 0.023 | 4810 (3160–6090) | <0.001 |
| D-dimer | 3.4 (1.7–7.8) | 8.1 (3.1–21) | 0.043 | 2.8 (2.5–6.6) | 0.50 |
| Lactate | 2.2 (1.7–2.6) | 2.6 (2.4–3.5) | 0.045 | 2.6 (2.3–3.4) | 0.043 |
| Creatinine | 95 (79–142) | 128 (100–218) | 0.047 | 121 (97–215) | 0.057 |
| Lowest $PaO_2/FiO_2$ ratio | 10.4 (9.6–13.2) | 9.4 (8.6–10.6) | 0.037 | 9.4 (7.6–12.3) | 0.11 |
| Mechanical ventilation | 29 (57%) | 7 (64%) | 0.68 | 10 (59%) | 0.89 |
| CRRT | 6 (12%) | 2 (18%) | 0.57 | 3 (18%) | 0.54 |
| Vasoactive treatment | 28 (55%) | 10 (91%) | 0.029 | 13 (77%) | 0.12 |

Data are presented as median (interquartile range) or number (percentages). Mann-Whitney U-test was used to compare continuous variables and $chi^2$-test was used to compare categorical variables between patients with normal ECG and prior MI pattern and ST-T pathology. MI: myocardial infarction, CRP: C-reactive protein, IL-6: Interleukin 6, NT-pro-BNP: N-Terminal pro brain natriuretic peptide, CRRT: Continuous renal replacement therapy. Laboratory reference ranges: CRP <5 mg/L; IL-6 <7,0 ng/L; Ferritin male patients 25–310 μg/L, female patients 10–155 μg/L (non-age-adjusted); Troponin I male patients <35 ng/L, female patients <16 ng/L; NT-pro-BNP male patients <230 ng/L, female patients <330 ng/L (non-age-adjusted); D-dimer <0.50 mg/L (non-age-adjusted); Lactate 0.5–1.6 mmol/L; Creatinine male patients 60–105 μmol/L, female patients 45–90 μmol/L.

with significant cardiovascular comorbidity that are at high risk for death independently of the disease severity of COVID-19 at ICU admission. On the contrary, ST-T pathology in this population may have several aetiologies, some of which may be a direct cause of COVID-19, such as myocarditis, right ventricular strain and demand ischemia due to hypoxemia [36–38]. ST-T pathology may thus be more likely to depend on the disease severity and consequently less likely to display a mortality association independent of SAPS 3 compared with prior MI pattern. Both chronic and acute ECG pathology at ICU admission was associated with higher peak values of biomarkers of cardiac strain and damage, lactate and more frequent requirement for vasoactive treatment. This indicates that ECG at ICU admission may be an important prognostic tool in COVID-19.

Similar to our study, a cohort study of 756 patients with COVID-19 found that prior MI pattern and T-wave inversion at hospital admission were associated with death while sinus tachycardia was not [21]. Further, patients with ST-T pathology at hospital admission have a higher risk of developing more severe disease [22]. One study of 850 patients with ECG recorded at presentation to the emergency department, and another study of 269 patients that analysed ECG at hospital admission and the seventh day of hospitalization found that ST-T-pathology is predictive of death and invasive ventilation [23, 39]. RBBB [21], AF [23], and LVH [39] at hospital admission also have been reported to indicate higher risk of death in patients with COVID-19. These abnormalities were uncommon in our study and therefore lacked sufficient statistical power for analysis. Contrary to our findings, a study describing ECG findings at hospital admission in 431 patients who later died or underwent invasive ventilation reported abnormal ECG in 93% of patients, a high prevalence of AF (22%) and signs of right ventricular strain (30%) [24]. In their cohort, patients were older (74 vs 61 years) and had higher over-all mortality (46% vs 28%) compared to our study, which may account for some of the differences between our study and theirs. Previous studies reported longer QT interval in patients with COVID-19 compared to patients without COVID-19 [40] and QT prolongation

after falling ill with COVID-19 compared to before [41]. Previously reported QT intervals ranges from a mean of 443–450 ms [40–43], which is somewhat longer than the mean of 440 ms reported in this study. This difference may at least in part be explained by that the present study includes younger patients and fewer patients with previous heart disease, both of which are risk factors for QT prolongation [40, 41, 44].

Both ECG pathology and elevated troponin have previously been associated with death [9]. There are several plausible explanations why severely ill COVID-19 patients develop ECG abnormalities and myocardial damage. Patients with pre-existing cardiovascular disease are more prone to develop secondary myocardial ischemia due to non-cardiac conditions [45] such as hypoxia [36]. Consistent with this, we found that abnormalities associated with prior MI was associated with death and developed higher peak troponin-I and NT-pro-BNP values compared to patients with normal ECG. Patients with ST-T pathology in our study also had higher odds of death and developed higher peak values of cardiac biomarkers compared to patients with normal ECG, which may be caused by several different pathophysiological mechanisms, such as ischemia due to pre-existing coronary stenosis with oxygen supply-demand mismatch [45], acute coronary syndrome due to plaque rupture secondary [19], myocardial microthrombi due to complement activation [37] or myocarditis [38]. T-wave inversion, present in 16% of our cohort, may be present in up to 57% of patients with myocarditis [46]. In a study of unselected patients recently recovered from COVID-19, 60% of patients had findings consistent with myocardial inflammation on magnetic resonance imaging [13]. Myocarditis may be non-viral [38], as a part of the hyperinflammatory response reported in COVID-19, but also due to direct viral infiltration in myocardial cells [12, 47, 48].

Although patients with COVID-19 have high risk of pulmonary embolism [49] and 11% of the patients in our cohort were diagnosed with pulmonary embolism, none of them had an ECG consistent with right ventricular strain. In a case series of 15 hospitalized patients with confirmed COVID-19 and pulmonary embolism, 33% had right ventricular strain pattern on ECG while two-thirds had non-specific ECG findings, such as sinus tachycardia [50]. The absence of right ventricular strain pattern in our study could simply be due to the absence of pulmonary embolism at the time of the ECG-recording. However, critically ill patients with COVID-19 related pulmonary embolism may also have less clot burden [51] compared to a general ICU population with pulmonary embolism, and right ventricular strain pattern may therefore not manifest on the ECG. Further, the incidence of pulmonary embolism was lower in our cohort than in other studies [52], possibly due to a higher dose low molecular weight heparin thromboprophylaxis at our ICU.

In a previous study, patients with an abnormal ECG developed higher peak plasma creatinine and had a higher incidence of continuous renal replacement therapy compared to patients with normal ECG [53]. In our study, peak plasma creatinine was similarly higher in patients with prior MI pattern compared to those with normal ECG. Patients with ST-T pathology also developed higher peak plasma creatinine values compared to patients with normal ECG, but this finding was only borderline significant, likely due to low statistical power. Kidney injury in COVID-19 is likely multifactorial and may be caused by several mechanisms, in part common to those responsible for cardiac injury, including both direct viral pathophysiological effects, systemic inflammation, hypovolemia and cardiopulmonary instability related to the degree of illness severity [2, 45, 54]. Pre-existing cardiovascular risk factors are frequent in COVID-19 patients [5] and may further contribute to the development of simultaneous cardiac and renal dysfunction.

Contrary to previous studies, where immune dysregulation has been proposed as a major mechanism for cardiac injury in COVID-19 [3] and cardiac biomarkers have been positively associated with inflammatory biomarkers [55, 56], we found no statistically significant difference in CRP, ferritin and IL-6, between patients with prior MI pattern or ST-T pathology compared

to patients with normal ECG in our study. Patients admitted to the ICU may represent a cohort of patients with severe inflammatory response regardless of myocardial injury which could explain the lack of difference in our study. Pathological ECG changes may thus not primarily be caused by more severe inflammation, but rather similar levels of inflammation causing ECG abnormalities and cardiac injury primarily in patients with pre-existing cardiac disease.

Strengths of this study included that ECG interpretation was conducted according to pre-specified criteria by two independent physicians blinded to patient outcomes. The study was conducted at a large tertiary referral centre with a large catchment area and all inhabitants in Sweden are covered by the public health insurance increasing generalizability of our study. Further, no patients were lost to follow up and there was minimal missing data.

There are also limitations of this study. The single centre design reduces generalizability and the small sample size hampers statistical power, especially in the multivariable analysis where only additional adjustment for SAPS 3 was feasible. A larger study would have allowed for adjustment for more covariates, thus reducing the risk of confounding [57]. Several patients did not have an ECG recorded at ICU admission, which may have led to selection bias if patients with pre-existing comorbidities or more severe disease were more likely to have an ECG recorded. However, the baseline characteristics of this sub-cohort were similar to the entire PronMed cohort of COVID-19 patients admitted to the ICU at Uppsala University Hospital [58]. The inclusion of patients who had an ECG recorded within 48 hours prior to and up to 72 hours after ICU admission may have led to comparison of ECGs recorded late, at a time of clinical decompensation to ECGs recorded early at a time of relatively less severe disease. However, patients who had ECG pathology had their ECGs recorded at similar time in relation to ICU admission, compared with patients who had normal ECGs. Furthermore, the results of the main analysis were robust in the sensitivity analysis which was restricted to patients with ECGs recorded within one day from ICU admission.

This study contains important new information for bedside clinicians and future studies. If confirmed, ECG findings presented herein may be used in prognostic tools for severe COVID-19 and the apparent lack of association between pathological ECG and inflammatory markers may further the understanding how COVID-19 affects the cardiovascular system.

## Conclusion

ECG indicative of both chronic and acute cardiac conditions at ICU admission due to severe COVID-19 were associated with higher mortality and higher levels of cardiac biomarkers. ECG is an invaluable low-risk investigation in a range of clinical scenarios. Our study suggests that an ECG provides important prognostic information in severe COVID-19 and should be considered in all critically ill COVID-19 patients.

## Supporting information

**S1 Appendix. Sensitivity analysis.**
(DOCX)

**S1 Checklist.**
(DOC)

## Acknowledgments

The authors thank Elin Söderman, Joanna Wessbergh, Labolina Spång, Erik Danielsson and Philip Karlsson for excellent technical and administrative assistance. We also thank the collaborators of the Uppsala Intensive Care COVID-19 research group:

Tomas Luther[1], Sara Bülow Anderberg[1], Anna Gradin[1], Sarah Galien[1], Sten Rubertsson[1] and Katja Hanslin[1]. Lead author: Robert Frithiof[1], robert.frithiof@surgsci.uu.se. Contributions: TS, SBA, AG, SG, SR and KH collected patient data

[1] Department of Surgical Sciences, Anaesthesiology and Intensive Care Medicine, Uppsala University, Uppsala, Sweden

## Author Contributions

**Conceptualization:** Michael Hultström.

**Formal analysis:** Jacob Rosén, Karl Stattin, Robert Frithiof.

**Investigation:** Jacob Rosén, Maria Noreland, Miklós Lipcsey, Robert Frithiof, Michael Hultström.

**Methodology:** Jacob Rosén, Maria Noreland, Karl Stattin, Miklós Lipcsey, Robert Frithiof, Andrei Malinovschi, Michael Hultström.

**Resources:** Robert Frithiof, Michael Hultström.

**Supervision:** Andrei Malinovschi, Michael Hultström.

**Visualization:** Jacob Rosén.

**Writing – original draft:** Jacob Rosén.

**Writing – review & editing:** Jacob Rosén, Maria Noreland, Karl Stattin, Miklós Lipcsey, Robert Frithiof, Andrei Malinovschi, Michael Hultström.

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
