## [Decision Letter · Decision Letter 0]

30 Oct 2021

PONE-D-21-28860ECG pathology and its association with death in critically ill COVID-19 patients, a cohort studyPLOS ONE

Dear Dr. Jacob Rosen,

Thank you for submitting your manuscript to PLOS ONE. After careful consideration, we feel that it has merit but does not fully meet PLOS ONE’s publication criteria as it currently stands. Therefore, we invite you to submit a revised version of the manuscript that addresses the points raised during the review process.

ACADEMIC EDITOR: Thank you very much for having submitted this paper to the journal for consideration. Although the topic is not particularly novel and some criticisms about the statistical method I would like to give you  the opportunity to try and improve the quality of our work. Besides all the reviewers' comments I'd like you to  discuss more about the QT interval in COVID 19 patients and in particular why the rate of prolongation was so low in your series and why it was not a associated to death.

We look forward to receiving your revised manuscript.

Kind regards,

Simone Savastano

Academic Editor

PLOS ONE

Additional Editor Comments (if provided):

Thank you very much for having submitted this paper to the journal for consideration. Although the topic is not particularly novel and some criticisms about the statistical method I would like to give you the opportunity to try and improve the quality of our work. Besides all the reviewers' comments I'd like you to discuss more about the QT interval in COVID 19 patients and in particular why the rate of prolongation was so low in your series and why it was not a associated to death.

Journal Requirements:

2. In ethics statement in the manuscript and in the online submission form, please provide additional information about the patient records/samples used in your retrospective study. Specifically, please ensure that you have discussed whether all data/samples were fully anonymized before you accessed them and/or whether the IRB or ethics committee waived the requirement for informed consent. If patients provided informed written consent to have data/samples from their medical records used in research, please include this information

“The study was funded by the SciLifeLab/KAW national COVID-19 research program project grant to MH (KAW 2020.0182), the Swedish Research Council grant to RF (2014-02569 and 2014-07606).”

We note that you have provided additional information within the Funding Section. Please note that funding information should not appear in other areas of your manuscript. We will only publish funding information present in the Funding Statement section of the online submission form.

“The study was funded by

    1. The SciLifeLab/Knut and Alice Wallenbergs foundations national COVID-19 research program project grant (https://kaw.wallenberg.org) to MH (grant number 2020.0182). The funders had no role in study design, data collection and analysis, decision to publish, or preparation of the manuscript.

     2. The Swedish Research Council grant (https://www.vr.se) to RF (grant numbers 2014-02569 and 2014-07606). The funders had no role in study design, data collection and analysis, decision to publish, or preparation of the manuscript.”

5. One of the noted authors is a group or consortium Uppsala Intensive Care COVID-19 Research Group. In addition to naming the author group, please list the individual authors and affiliations within this group in the acknowledgments section of your manuscript. Please also indicate clearly a lead author for this group along with a contact email address

6. We note that you have indicated that data from this study are available upon request. PLOS only allows data to be available upon request if there are legal or ethical restrictions on sharing data publicly. For more information on unacceptable data access restrictions, please see http://journals.plos.org/plosone/s/data-availability#loc-unacceptable-data-access-restrictions.

Reviewers' comments:

Reviewer's Responses to Questions

**Comments to the Author**

1. Is the manuscript technically sound, and do the data support the conclusions?

Reviewer #1: No

Reviewer #2: Yes

2. Has the statistical analysis been performed appropriately and rigorously? 

Reviewer #1: No

Reviewer #2: Yes

3. Have the authors made all data underlying the findings in their manuscript fully available?

Reviewer #1: No

Reviewer #2: No

4. Is the manuscript presented in an intelligible fashion and written in standard English?

Reviewer #1: Yes

Reviewer #2: Yes

5. Review Comments to the Author

Reviewer #1: PONE-D-21-28860: statistical review

SUMMARY. This is a cohort study that focuses on the association between ECG abnormalities and total mortality. The statistical analysis relies on a battery of univariate and multivariate logistic regressions, followed by the computation of survival curves that compare subjects with and without ECG abnormalities. My major concern with this paper is that not all the available information has been exploited in both the logistics regression analysis (see major issue 1) and in the survival analysis (major issue 2). I futhermore list below some specific points that should be addressed.

MAJOR ISSUES

1) Table 3 displays the outputs of univariate logistic regressions where single and composite ECG abnormalities are separately included and the output of (bivariate) logistics regressions where the effect of composite abnormalities is adjusted for SAPS 3. Why are all the available covariates (described in Table 1) not included in the analysis? Ignoring the observed heterogeneity of the sample could bias the final results. Either the authors should motivate this choice, or they should include such information. In the latter case, given the limited sample size, a careful model selection procedure should be implemented, in order to include only the relevant covariates.

2) Survival analysis reduces to comparing the survival curves of subjects with and without ECG abnormalities. Again, why are the additional covariates not included in the analysis? Mortality risks should be adjusted for the variables of Table 1. The natural approach would rely on the estimation of a Cox regression model, which would be much more informative than the plots displayed by Figures 1 and 2.

SPECIFIC ISSUES

1) Page 4: subjects were included "in this study if they had an ECG recorded within 48 hours prior to, or within 72 hours after ICU admission". Could the authors clarify that they are not introducing selection bias with this choice? Is the final sample still a random subset of the target population?

2) I understand that the study aims at investigating the effects of ECG abnormalities within COVID subjects. However, shouldn't the study compare subjects with covid and without covid? Under this design, we could test whether ECG abnormalities interact with Covid and see whether ECG and covid influence additively or multiplicatively mortality risk.

3) Figures 1 and 2 should include the confidence bands of the survival curves. In addition, please specify the test that has been run to compare the curves (log-ranks test?).

Reviewer #2: Thank you for the opportunity to review your manuscript entitled "ECG pathology and its association with death in critically ill COVID-19 patients, a cohort study". First of all, I would like to suggest you specify the total word count at the beginning of the manuscript and insert line numbers to make it easier to review and to correct any mistake. Then, you can find my specific comments here below.

METHODS

-Which is the rationale for including patients with an ECG recorded in the time interval between 48 hours before and 72 hours after ICU admission, instead of patients with an ECG recorded at ICU admission or patients admitted to ICU with an ECG recorded at hospital admission? In 5 hours, it is not likely that a patient will undergo pharmacological treatments or procedures which can modify ECG pattern, is it? Furthermore, in case of more than one ECG recorded, which one did you consider for the analysis? Please, specify these details.

-Which pre-specified criteria were used to analyse ECG? Please, explain.

-Why did you exclude the criteria for LVH, RVH and conduction blocks among the ECGs with prior myocardial infarction pattern? These conditions could co-exist and the firsts do not exclude the latters. Consider changing this definition, which moreover contrasts with the one written in the caption of Table 2 (see comment below in Results section).

RESULTS

-Consider adding a figure with a flow chart to clarify the inclusion and exclusion criteria of the study population.

-In Table 1 “Patient characteristics at baseline and ICU admission” you miss the percentage symbol in the cell about the females among all patients.

-As mentioned in the previous comment in Methods section, in the Table 2 caption you rightly defined prior myocardial infarction as ECG with Q-wave and/or poor R-wave progression. Please, correct the definition used in the Methods section, which is different.

-In Table 3 remove the percentage symbol in the cell about the heart rate among dead patients (the number in the brackets should indicate the standard deviation).

-Why did you use NT-proBNP and not BNP? The results could be influenced by renal dysfunction, which should be taken into consideration when interpreting these data. Circulating levels of both BNP and NT-proBNP increase indeed with kidney failure, but the impact of kidney function on NT-proBNP is much more pronounced than that on BNP (Takase H, Dohi Y. Kidney function crucially affects B-type natriuretic peptide (BNP), N-terminal proBNP and their relationship. Eur J Clin Invest. 2014;44(3):303-8. doi: 10.1111/eci.12234. Epub 2014 Jan 20. PMID: 24372567).

DISCUSSION

-Citing the paper of Bertini M et al. (Bertini M, Ferrari R, Guardigli G, et al. Electrocardiographic features of 431 consecutive, critically ill COVID-19 patients: an insight into the mechanisms of cardiac involvement. Europace. Epub ahead of print 18 September 2020. DOI: 10.1093/europace/euaa258), you wrote that you reported the proportion of patients who were not eligible for ICU admission, unlike them. However, I have not found this information throughout your manuscript. You should add your criteria of ICU admission in the Methods section, before indicating the inclusion criteria of the study population.

6. PLOS authors have the option to publish the peer review history of their article (what does this mean?). If published, this will include your full peer review and any attached files.

Reviewer #1: No

Reviewer #2: No

---

## [Author Response · Author response to Decision Letter 0]

12 Nov 2021

Authors’ response to reviewers (also provided as a word file)

Additional Editor Comments (if provided):

Thank you very much for having submitted this paper to the journal for consideration. Although the topic is not particularly novel and some criticisms about the statistical method I would like to give you the opportunity to try and improve the quality of our work. Besides all the reviewers' comments I'd like you to discuss more about the QT interval in COVID 19 patients and in particular why the rate of prolongation was so low in your series and why it was not a associated to death.

Thank you for the opportunity to revise our manuscript. We have made several alterations according to the reviewers’ comments, which we believe have greatly improved our manuscript. References are found at the end of this response letter. References to page and line numbers are 

QT prolongation has been widely reported among patients with COVID. In our study, the mean QTc, corrected according to Bazett, was 440 ms and two patients (3%) had QTc >500 ms. The mean QTc in our study is comparable, but often marginally shorter, than figures reported by Rubin[1] (450 ms), Akthar[2] (461 and 449 ms), Changal[3] (446 ms), Garcia-Rodriguez[4] (443 ms). This incongruity may be due to differences in patient characteristics, as our cohort is younger and have fewer cardiovascular comorbidities than patients in previous studies, both of which are risk factors for prolonged QTc.

The difference in QTc among patients who survived (441 ms) and patients who died (437 ms) was very small, and a lack of association with death may be due to lack of statistical power or due to our younger cohort of patients. We have added a paragraph to the manuscript discussing this (p 15, lines 278-284). 

Journal Requirements:

The manuscript has been altered to adhere to PLOS ONE style requirements.

2. In ethics statement in the manuscript and in the online submission form, please provide additional information about the patient records/samples used in your retrospective study. Specifically, please ensure that you have discussed whether all data/samples were fully anonymized before you accessed them and/or whether the IRB or ethics committee waived the requirement for informed consent. If patients provided informed written consent to have data/samples from their medical records used in research, please include this information

Patients provided written informed consent. If consent was not possible to obtain from the patient, written informed consent was provided by the patients’ next of kin, as per decision from the ethics committee. Data was collected from electronic patient records, and as such anonymization before collection was not possible. Collected data was entered into a pseudoanynymized dataset and all analyses were performed on pseudoanonymized data. This has been clarified in the methods section (p 4, lines 80-86). 

“The study was funded by the SciLifeLab/KAW national COVID-19 research program project grant to MH (KAW 2020.0182), the Swedish Research Council grant to RF (2014-02569 and 2014-07606).”

We note that you have provided additional information within the Funding Section. Please note that funding information should not appear in other areas of your manuscript. We will only publish funding information present in the Funding Statement section of the online submission form.

“The study was funded by

 1. The SciLifeLab/Knut and Alice Wallenbergs foundations national COVID-19 research program project grant (https://kaw.wallenberg.org) to MH (grant number 2020.0182). The funders had no role in study design, data collection and analysis, decision to publish, or preparation of the manuscript.

 2. The Swedish Research Council grant (https://www.vr.se) to RF (grant numbers 2014-02569 and 2014-07606). The funders had no role in study design, data collection and analysis, decision to publish, or preparation of the manuscript.”

The manuscript has been changed accordingly. An amended Funding Statement is included in the cover letter.

Unfortunately, it is not possible to upload the study’s minimal underlying dataset. Data privacy regulations prohibit deposition of individual level data to public repositories and the ethical approval does not cover public sharing of data for unknown purposes. Upon contact with the authors or SciLifeLab (https://doi.org/10.17044/scilifelab.14229410.v1) an institutional data transfer agreement may be established, and data shared if the aims of data use are covered by ethical approval and patient consent.

5. One of the noted authors is a group or consortium Uppsala Intensive Care COVID-19 Research Group. In addition to naming the author group, please list the individual authors and affiliations within this group in the acknowledgments section of your manuscript. Please also indicate clearly a lead author for this group along with a contact email address

The Acknowledgements section has been edited accordingly.

6. We note that you have indicated that data from this study are available upon request. PLOS only allows data to be available upon request if there are legal or ethical restrictions on sharing data publicly. For more information on unacceptable data access restrictions, please see http://journals.plos.org/plosone/s/data-availability#loc-unacceptable-data-access-restrictions.

Unfortunately, it is not possible to upload the study’s minimal underlying dataset. Data privacy regulations prohibit deposition of individual level data to public repositories and the ethical approval does not cover public sharing of data for unknown purposes. Upon contact with the authors or SciLifeLab (https://doi.org/10.17044/scilifelab.14229410.v1) an institutional data transfer agreement may be established, and data shared if the aims of data use are covered by ethical approval and patient consent.

Reviewers' comments:

Reviewer's Responses to Questions

Comments to the Author

1. Is the manuscript technically sound, and do the data support the conclusions?

Reviewer #1: No

Reviewer #2: Yes

2. Has the statistical analysis been performed appropriately and rigorously?

Reviewer #1: No

Reviewer #2: Yes

3. Have the authors made all data underlying the findings in their manuscript fully available?

Reviewer #1: No

Reviewer #2: No

4. Is the manuscript presented in an intelligible fashion and written in standard English?

Reviewer #1: Yes

Reviewer #2: Yes

5. Review Comments to the Author

Reviewer #1: PONE-D-21-28860: statistical review

SUMMARY. This is a cohort study that focuses on the association between ECG abnormalities and total mortality. The statistical analysis relies on a battery of univariate and multivariate logistic regressions, followed by the computation of survival curves that compare subjects with and without ECG abnormalities. My major concern with this paper is that not all the available information has been exploited in both the logistics regression analysis (see major issue 1) and in the survival analysis (major issue 2). I futhermore list below some specific points that should be addressed.

Thank you for your time and expertise reviewing our manuscript. We hope our responses, alterations to the manuscript and new analyses are satisfactory. References are found at the end of this response letter. 

MAJOR ISSUES

1) Table 3 displays the outputs of univariate logistic regressions where single and composite ECG abnormalities are separately included and the output of (bivariate) logistics regressions where the effect of composite abnormalities is adjusted for SAPS 3. Why are all the available covariates (described in Table 1) not included in the analysis? Ignoring the observed heterogeneity of the sample could bias the final results. Either the authors should motivate this choice, or they should include such information. In the latter case, given the limited sample size, a careful model selection procedure should be implemented, in order to include only the relevant covariates.

Not all available covariates are included in the analysis, as our study unfortunately was not powered to do so. Including more than one variable per ten events in a multivariable model may introduce bias.[5] Our study includes 22 deaths, and hence inclusion of more than two variables may be inappropriate. Considering this limitation, we believe SAPS 3 to be the best variable to adjust for baseline differences, as it is a well-validated scoring system used for prediction of hospital mortality based on several variables of physiological derangements, current and previous conditions. The rationale for using SAPS 3 as adjusting covariate is added to the methods section and the discussion (p 6, lines 136-138) has been edited to emphasize this limitation (p 17, 345-346). 

2) Survival analysis reduces to comparing the survival curves of subjects with and without ECG abnormalities. Again, why are the additional covariates not included in the analysis? Mortality risks should be adjusted for the variables of Table 1. The natural approach would rely on the estimation of a Cox regression model, which would be much more informative than the plots displayed by Figures 1 and 2.

The Kaplan-Meier curves illustrate mortality in patients with normal ECG compared to patients with prior MI pattern and ST-T pathology respectively. The results section was not well worded and has been edited for clarity.

This analysis is also limited by the study size and the number of events (deaths). The 22 deaths in the study only reliably permits the inclusion of two variables in the adjusted model. As the time perspective is rather short, reporting 30-day mortality, we chose to perform a logistic regression for odds of death instead of a Cox regression. The Kaplan-Meier plots are an unadjusted illustration detailing the timing of the deaths, whereas the adjusted logistic regression attempts to control for potential confounding to the extent permitted by the study size. 

SPECIFIC ISSUES

1) Page 4: subjects were included "in this study if they had an ECG recorded within 48 hours prior to, or within 72 hours after ICU admission". Could the authors clarify that they are not introducing selection bias with this choice? Is the final sample still a random subset of the target population?

Good point. Knowledge of prior heart or vascular disease, or risk factors thereof, may have alerted the clinician to the risk of current cardiac involvement and prompted ECG recording. Thus, this study may include more patients with heart disease than other cohorts of COVID patients. However, this subcohort did not differ substantially in patient characteristics compared to the entire PronMed cohort[6,7] of COVID patients admitted to the ICU at Uppsala University Hospital (not limited to only patients with an ECG recorded). Further, this would not affect the internal validity of the results. 

We chose a time interval restriction because the aim of the study was to compare the prevalence of ECG abnormalities in COVID patients at ICU admission and their association with mortality, as we thought this association would be interesting to bedside clinicians. However, we agree that inclusion of patients who had an ECG recorded within 48 hours prior to and up to 72 hours after ICU admission may have led to comparison of ECGs recorded late, at a time of clinical decompensation to ECGs recorded early at a time of relatively less severe disease. However, patients who had ECG pathology had their ECGs recorded at similar time in relation to ICU admission (information added to Table 3), compared with patients who had normal ECGs. Furthermore, the results of the main analysis were robust in a sensitivity analysis which was restricted to patients with ECGs recorded within one day from ICU admission. The sensitivity analysis was added as separate file (S1 Appendix) and the limitations paragraph has been edited to include this weakness (p 17-18, lines 348-357).

2) I understand that the study aims at investigating the effects of ECG abnormalities within COVID subjects. However, shouldn't the study compare subjects with covid and without covid? Under this design, we could test whether ECG abnormalities interact with Covid and see whether ECG and covid influence additively or multiplicatively mortality risk.

This would indeed be a very interesting analysis! However, the aim of the study was to investigate ECG abnormalities and their association with death in COVID patients. It is beyond the scope of the present study to also investigate how ECG abnormalities interact with different diseases.

3) Figures 1 and 2 should include the confidence bands of the survival curves. In addition, please specify the test that has been run to compare the curves (log-ranks test?).

Thank you, this would improve the figures. Confidence bands have been added and the use of log rank test specified.

Reviewer #2: Thank you for the opportunity to review your manuscript entitled "ECG pathology and its association with death in critically ill COVID-19 patients, a cohort study". First of all, I would like to suggest you specify the total word count at the beginning of the manuscript and insert line numbers to make it easier to review and to correct any mistake. Then, you can find my specific comments here below.

Thank you for reviewing our manuscript. It has been changed accordingly. References are found at the end of this response letter. 

METHODS

-Which is the rationale for including patients with an ECG recorded in the time interval between 48 hours before and 72 hours after ICU admission, instead of patients with an ECG recorded at ICU admission or patients admitted to ICU with an ECG recorded at hospital admission? In 5 hours, it is not likely that a patient will undergo pharmacological treatments or procedures which can modify ECG pattern, is it? Furthermore, in case of more than one ECG recorded, which one did you consider for the analysis? Please, specify these details.

ECG at hospital admission has previously been studied, and abnormalities are associated with increased mortality. We wanted to specifically study patients admitted to the ICU, as this is a group with higher mortality. Furthermore, patients are admitted to the ICU at a time of clinical deterioration, which may reveal other ECG patterns than ECG recorded at hospital admission. Many patients were admitted to the hospital several days prior to ICU admission. In order to collect and compare ECGs at ICU admission, we had to arbitrarily define time constraints.

However, we agree that inclusion of patients who had an ECG recorded within 48 hours prior to and up to 72 hours after ICU admission may have led to comparison of ECGs recorded late, at a time of clinical decompensation to ECGs recorded early at a time of relatively less severe disease. However, patients who had ECG pathology had their ECGs recorded at similar time in relation to ICU admission (information added to Table 3), compared with patients who had normal ECGs. Furthermore, the results of the main analysis were robust in a sensitivity analysis which was restricted to patients with ECGs recorded within one day from ICU admission. The sensitivity analysis was added as separate file (S1 Appendix) and the limitations paragraph has been edited to include this weakness (p 17-18, lines 348-357).

-Which pre-specified criteria were used to analyse ECG? Please, explain.

We used criteria that adhered to the published guidelines[8–11]. The interpretation data were entered in a pre-defined case report form. 

-Why did you exclude the criteria for LVH, RVH and conduction blocks among the ECGs with prior myocardial infarction pattern? These conditions could co-exist and the firsts do not exclude the latters. Consider changing this definition, which moreover contrasts with the one written in the caption of Table 2 (see comment below in Results section).

This section was poorly phrased. We only excluded these patterns if it was obvious that poor R-wave progression was due to one of these conditions. The diagnosis of myocardial infarction based on poor R-wave progression on surface-ECG in the prescence of LVH, RVH and conduction block may be difficult or impossible. We therefore did not include ECG with poor R-wave progression that was obviously due to other conditions than myocardial infarction in this group to increase specificity, as demonstrated by Zema et al.[12] We have rephrased this to clarify (p 5, lines 123-126). 

RESULTS

-Consider adding a figure with a flow chart to clarify the inclusion and exclusion criteria of the study population.

A flow chart has been added as Fig 1. 

-In Table 1 “Patient characteristics at baseline and ICU admission” you miss the percentage symbol in the cell about the females among all patients.

Thank you, Table 1 has been corrected.

-As mentioned in the previous comment in Methods section, in the Table 2 caption you rightly defined prior myocardial infarction as ECG with Q-wave and/or poor R-wave progression. Please, correct the definition used in the Methods section, which is different.

See comment concerning ECG patterns above. The text has been altered for consistency.

-In Table 3 remove the percentage symbol in the cell about the heart rate among dead patients (the number in the brackets should indicate the standard deviation).

It has been removed.

-Why did you use NT-proBNP and not BNP? The results could be influenced by renal dysfunction, which should be taken into consideration when interpreting these data. Circulating levels of both BNP and NT-proBNP increase indeed with kidney failure, but the impact of kidney function on NT-proBNP is much more pronounced than that on BNP (Takase H, Dohi Y. Kidney function crucially affects B-type natriuretic peptide (BNP), N-terminal proBNP and their relationship. Eur J Clin Invest. 2014;44(3):303-8. doi: 10.1111/eci.12234. Epub 2014 Jan 20. PMID: 24372567).

We display NT-proBNP as it is the standard analysis at our institution. Although listing BNP was not an alternative, we agree that it may have reduced the risk of confounding by kidney function.

DISCUSSION

-Citing the paper of Bertini M et al. (Bertini M, Ferrari R, Guardigli G, et al. Electrocardiographic features of 431 consecutive, critically ill COVID-19 patients: an insight into the mechanisms of cardiac involvement. Europace. Epub ahead of print 18 September 2020. DOI: 10.1093/europace/euaa258), you wrote that you reported the proportion of patients who were not eligible for ICU admission, unlike them. However, I have not found this information throughout your manuscript. You should add your criteria of ICU admission in the Methods section, before indicating the inclusion criteria of the study population.

This section was poorly worded and has been edited for clarity. We do indeed not report which patients were ineligible for ICU admission. Thank you for your careful reading. Patients were admitted to ICU based on the clinical judgement of the attending ICU physician. The main criterium was need for organ-support, most commonly need for respiratory support beyond high-flow nasal oxygen with 60% oxygen at 60L min-1, or high risk of progression based on clinical judgement. The methods section has been edited (p 4-5, lines 100-103). 

6. PLOS authors have the option to publish the peer review history of their article (what does this mean?). If published, this will include your full peer review and any attached files.

Do you want your identity to be public for this peer review? For information about this choice, including consent withdrawal, please see our Privacy Policy.

Reviewer #1: No

Reviewer #2: No

References

1. Rubin GA, Desai AD, Chai Z, Wang A, Chen Q, Wang AS, et al. Cardiac Corrected QT Interval Changes Among Patients Treated for COVID-19 Infection During the Early Phase of the Pandemic. JAMA Network Open. 2021;4: e216842. doi:10.1001/jamanetworkopen.2021.6842

2. Akhtar Z, Gallagher MM, Yap YG, Leung LWM, Elbatran AI, Madden B, et al. Prolonged QT predicts prognosis in COVID-19. Pacing Clin Electrophysiol. 2021;44: 875–882. doi:10.1111/pace.14232

3. Changal K, Paternite D, Mack S, Veria S, Bashir R, Patel M, et al. Coronavirus disease 2019 (COVID-19) and QTc prolongation. BMC Cardiovascular Disorders. 2021;21: 158. doi:10.1186/s12872-021-01963-1

4. García-Rodríguez D, Remior P, García-Izquierdo E, Toquero J, Castro V, Fernández Lozano I. Drug-induced QT prolongation in COVID-19 pneumonia: influence on in-hospital survival. Rev Esp Cardiol. 2021;74: 111–112. doi:10.1016/j.rec.2020.09.027

5. Peduzzi P, Concato J, Kemper E, Holford TR, Feinstein AR. A simulation study of the number of events per variable in logistic regression analysis. Journal of Clinical Epidemiology. 1996;49: 1373–1379. doi:10.1016/S0895-4356(96)00236-3

6. PRONMED Uppsala COVID-19 ICU Biobank. SciLifeLab; 2021. doi:10.17044/scilifelab.14229410.v1

7. Sancho Ferrando E, Hanslin K, Hultström M, Larsson A, Frithiof R, Lipcsey M, et al. Soluble TNF receptors predict acute kidney injury and mortality in critically ill COVID-19 patients: A prospective observational study. Cytokine. 2021;149: 155727. doi:10.1016/j.cyto.2021.155727

8. Surawicz B, Childers R, Deal BJ, Gettes LS, Bailey JJ, Gorgels A, et al. AHA/ACCF/HRS recommendations for the standardization and interpretation of the electrocardiogram: part III: intraventricular conduction disturbances: a scientific statement from the American Heart Association Electrocardiography and Arrhythmias Committee, Council on Clinical Cardiology; the American College of Cardiology Foundation; and the Heart Rhythm Society. Endorsed by the International Society for Computerized Electrocardiology. J Am Coll Cardiol. 2009;53: 976–981. doi:10.1016/j.jacc.2008.12.013

9. Rautaharju PM, Surawicz B, Gettes LS. AHA/ACCF/HRS Recommendations for the Standardization and Interpretation of the Electrocardiogram: Part IV: The ST Segment, T and U Waves, and the QT Interval A Scientific Statement From the American Heart Association Electrocardiography and Arrhythmias Committee, Council on Clinical Cardiology; the American College of Cardiology Foundation; and the Heart Rhythm Society Endorsed by the International Society for Computerized Electrocardiology. Journal of the American College of Cardiology. 2009;53: 982–991. doi:10.1016/j.jacc.2008.12.014

10. Hancock EW, Deal BJ, Mirvis DM, Okin P, Kligfield P, Gettes LS, et al. AHA/ACCF/HRS recommendations for the standardization and interpretation of the electrocardiogram: part V: electrocardiogram changes associated with cardiac chamber hypertrophy: a scientific statement from the American Heart Association Electrocardiography and Arrhythmias Committee, Council on Clinical Cardiology; the American College of Cardiology Foundation; and the Heart Rhythm Society. Endorsed by the International Society for Computerized Electrocardiology. J Am Coll Cardiol. 2009;53: 992–1002. doi:10.1016/j.jacc.2008.12.015

11. Wagner GS, Macfarlane P, Wellens H, Josephson M, Gorgels A, Mirvis DM, et al. AHA/ACCF/HRS recommendations for the standardization and interpretation of the electrocardiogram: part VI: acute ischemia/infarction: a scientific statement from the American Heart Association Electrocardiography and Arrhythmias Committee, Council on Clinical Cardiology; the American College of Cardiology Foundation; and the Heart Rhythm Society. Endorsed by the International Society for Computerized Electrocardiology. J Am Coll Cardiol. 2009;53: 1003–1011. doi:10.1016/j.jacc.2008.12.016

12. Zema MJ, Collins M, Alonso DR, Kligfield P. Electrocardiographic Poor R-Wave Progression: Correlation with Postmortem Findings. CHEST. 1981;79: 195–200. doi:10.1378/chest.79.2.195

---

## [Decision Letter · Decision Letter 1]

1 Dec 2021

ECG pathology and its association with death in critically ill COVID-19 patients, a cohort study

PONE-D-21-28860R1

Dear Dr. Jacob Rosen,

We’re pleased to inform you that your manuscript has been judged scientifically suitable for publication and will be formally accepted for publication once it meets all outstanding technical requirements.

Kind regards,

Simone Savastano

Academic Editor

PLOS ONE

Additional Editor Comments (optional):

Thank you very much for having addressed all the comments of the reviewers.

Reviewers' comments:

Reviewer's Responses to Questions

**Comments to the Author**

1. If the authors have adequately addressed your comments raised in a previous round of review and you feel that this manuscript is now acceptable for publication, you may indicate that here to bypass the “Comments to the Author” section, enter your conflict of interest statement in the “Confidential to Editor” section, and submit your "Accept" recommendation.

Reviewer #1: All comments have been addressed

Reviewer #2: All comments have been addressed

2. Is the manuscript technically sound, and do the data support the conclusions?

Reviewer #1: (No Response)

Reviewer #2: (No Response)

3. Has the statistical analysis been performed appropriately and rigorously? 

Reviewer #1: (No Response)

Reviewer #2: (No Response)

4. Have the authors made all data underlying the findings in their manuscript fully available?

Reviewer #1: (No Response)

Reviewer #2: (No Response)

5. Is the manuscript presented in an intelligible fashion and written in standard English?

Reviewer #1: (No Response)

Reviewer #2: (No Response)

6. Review Comments to the Author

Reviewer #1: (No Response)

Reviewer #2: (No Response)

7. PLOS authors have the option to publish the peer review history of their article (what does this mean?). If published, this will include your full peer review and any attached files.

Reviewer #1: No

Reviewer #2: No

---

## [Editor Report · Acceptance letter]

6 Dec 2021

PONE-D-21-28860R1 

ECG pathology and its association with death in critically ill COVID-19 patients, a cohort study. 

Dear Dr. Rosén:

I'm pleased to inform you that your manuscript has been deemed suitable for publication in PLOS ONE. Congratulations! Your manuscript is now with our production department. 

Kind regards, 

on behalf of

Dr. Simone Savastano 

Academic Editor

PLOS ONE